# Metabolic Outcomes in Southern Italian Preadolescents Residing Near an Industrial Complex: The Role of Residential Location and Socioeconomic Status

**DOI:** 10.3390/ijerph16112036

**Published:** 2019-06-08

**Authors:** Esha Bansal, Donatella Placidi, Shaye Carver, Stefano Renzetti, Augusto Giorgino, Giuseppa Cagna, Silvia Zoni, Chiara Fedrighi, Miriana Montemurro, Manuela Oppini, Michele Conversano, Stefano Guazzetti, Robert O. Wright, Donald Smith, Luz Claudio, Roberto G. Lucchini

**Affiliations:** 1Department of Medical and Surgical Specialties, Radiological Sciences and Public Health, University of Brescia, 25123 Brescia, Italy; esha.bansal@icahn.mssm.edu (E.B.); carversh@bc.edu (S.C.); giuseppa.cagna@unibs.it (G.C.); silvia.zoni@unibs.it (S.Z.); chiara.fedrighi@unibs.it (C.F.); miriana.montemurro@unibs.it (M.M.); manuela.oppini@unibs.it (M.O.); stefano.guazzetti@ausl.re.it (S.G.);roberto.lucchini@unibs.it (R.G.L.); 2Department of Environmental Medicine and Public Health, Icahn School of Medicine at Mount Sinai, New York, NY 10029, USA; robert.wright@mssm.edu (R.O.W.); luz.claudio@mssm.edu (L.C.); 3Department of Biology, Morrissey College of Arts and Sciences, Boston College, Chestnut Hill, PA 02467, USA; 4Department of Molecular and Translational Medicine, University of Brescia, 25123 Brescia, Italy; stefano.renzetti@unibs.it; 5Department of Clinical Sciences and Community Health, University of Milan, 20122 Milan, Italy; 6Department of Prevention, Local Health Authority of Taranto, 74121 Taranto, Italy; augusto.giorgino@asl.taranto.it (A.G.); michele.conversano@asl.taranto.it (M.C.); 7Local Health Authority of Reggio Emilia, 42122 Reggio Emilia, Italy; 8Department of Microbiology and Environmental Toxicology, University of California Santa Cruz, Santa Cruz, CA 95064, USA; drsmith@ucsc.edu

**Keywords:** children, socioeconomic status, residential location, industrial, air pollution, body mass index, BMI, blood glucose, obesity

## Abstract

Evidence suggests that environmental exposures and socioeconomic factors may interact to produce metabolic changes in children. We assessed the influence of residential location and socioeconomic status (SES) on pediatric body mass index (BMI) Z-score and fasting blood glucose (FBG) concentration. Participants included 214 children aged 6–11 years who live near a large industrial complex in Taranto, Italy. Participants were grouped into residential zones based on the distance between their home address and the complex periphery (Zone 1: 0.000–4.999 km, Zone 2: 5.000–9.999 km, Zone 3: 10.000–15.000 km). BMI Z-scores were calculated via World Health Organization (WHO) pediatric reference curves. FBG was obtained via venous blood sampling. Closer residential location to the industrial complex on the order of 5.000 km was significantly associated with worsened metabolic outcomes, particularly in female children. Zone 1 participants had higher BMI-adjusted FBG than Zone 2 and 3 participants (*p* < 0.05 versus Zone 2; *p* < 0.01 versus Zone 3). SES did not significantly influence BMI-adjusted FBG. Moreover, BMI Z-scores indicated high rates of overweight (22.0%) and obesity (22.9%) in the cohort. BMI Z-score was not significantly associated with SES or residential zone but was negatively associated with maternal education level (*p* < 0.05). These results offer new evidence that residing near industrial activity may predict adverse effects on child metabolic health.

## 1. Introduction

Exposure to industrial pollutants is a serious environmental health problem, particularly as it affects children. Even low-level toxicity from contaminated air, water, and food sources has been implicated in pediatric pathologies of high morbidity and mortality, such as obesity, organ damage, and neurological dysfunction [1,2,3].

Many of the pediatric metabolic conditions linked to environmental exposure have been studied and found to be associated with specific sociodemographic conditions. This is particularly true for the Mediterranean region of Europe, where childhood obesity is increasing at alarming rates [4,5,6,7]. In Italy, 22.2% of children at the age of 8–9 years are overweight, and 17.4% are obese. These rates have worsened substantially over the past decades, particularly for girls, yet remain higher in males relative to females and in Southern relative to Northern Italy [4,7,8]. A recent study of 15,035 Italian schoolchildren aged 11–15 years found 21.8% of boys and 13.3% of girls to be overweight or obese in 2014 [9]. These findings agree with prior literature, which associates childhood obesity with low socioeconomic status (SES) and low maternal education level [5,8,9,10]. Childhood obesity has also been linked to cognitive conditions, including ADHD, risky behavior, and antisocial behavior [11,12].

However, such health conditions have not been as well described in the context of multiple sociodemographic and anthropometric variables, or in the setting of industrial activity. Much remains to be understood about the role of sociodemographic and lifestyle factors in mediating harmful pediatric health effects linked to environmental exposure. Numerous chemicals present in industrial sites—including heavy metals, polychlorinated biphenyls, dioxins, and bisphenol A—are considered obesogens because they have been shown to alter adipose tissue formation and neuroendocrine regulation of appetite [13]. Exposure to industrial chemicals may also increase obesity indirectly by promoting impulsive behaviors that interfere with appetite control [12]. While children are far more sensitive than adults to such toxins, relatively little is known about the differential effects of each chemical based on child characteristics such as sex and body size [14]. To our knowledge, there is also scarce currently published work that evaluates the metabolic outcomes of children living near industrial sites. This knowledge gap could limit the investigation of environmental contributors to metabolic syndromes. 

In addition to body size metrics, metabolic pathologies have been described in relation to early-life environmental exposure. For instance, the ambient air pollutants ozone and sulfate have been characterized as triggers of Type I diabetes in early childhood [15]. Literature indicates that arsenic, a heavy metal compound produced by steel plants, contributes to both obesity and prediabetic insulin resistance, an important step in the pathogenesis of Type II diabetes [16].

This study was conducted in the areas surrounding an industrial complex that includes the largest steel plant in the European Union (Taranto, Italy). The industrial activities of this complex are known to produce substances that affect child obesity and metabolism. The objective of this study is to describe the metabolic health effects of industrial site proximity on school-aged children, a population segment that has not been well studied in this context. In particular, this paper aims to: (1) describe trends in body mass index (BMI) Z-score and fasting blood glucose (FBG) concentration among a cohort of urban children aged 6–11 years residing near the industrial complex; and (2) relate these findings to children’s residential distance from the industrial complex and SES. Our approach combines bodily measurements and physiological data with sociodemographic and geographic information. In this way, the study offers a deeper analysis of the relationship between metabolic outcomes and environmental exposure than is typically possible in pediatric population studies about industrial activity. The authors intend that these analyses will be considered when designing future health interventions for children with environmental exposure to industrial emissions. 

## 2. Materials and Methods

### 2.1. Study Design and Population 

A cross-sectional study was carried out with a cohort of 214 primary school children aged 6–11 years, recruited from 12 public primary schools in the Taranto municipality. Among the 432 children who received parental consent to participate in this study, full or partial data were collected from the 312 children that satisfied the inclusion and exclusion criteria. Of the 312 eligible participants, 214 participants (68.6%) provided sufficient data to be included in the analyses of this paper. The cohort of 214 participants included in these analyses did not differ significantly from the 98 participants whose data were not used, on any analyzed characteristic. A more detailed breakdown of enrollment data is available upon request. 

*Inclusion Criteria:* To participate in the study: (1) children must have been born and raised continuously in one of the target study areas in the province of Taranto, Italy; and (2) the pregnancy of the child’s mother must have been carried out within the participant’s current area of residence at the time of recruitment.

*Exclusion Criteria:* Subjects were excluded from the study for having one or more of the following conditions: history of total parenteral nutrition; history of neurodegenerative disease; current use of pharmaceuticals affecting the nervous system or with known neuropsychological side effects; inadequately corrected visual defects; deficits in hand and/or finger function; and/or presence of a neurological, neuropsychiatric, hematological, metabolic, endocrine, kidney, or bile tract pathology. Children excluded due to pathologies were referred to the pediatric care service of the Azienda Sanitaria Locale (ASL) of Taranto.

Because participants were below 12 years of age, informed written consent for child participation and for anonymous data collection was obtained from both parents of each participant. Children were enrolled in this study from March 2015 to May 2016. This study was conducted in accordance with the Declaration of Helsinki. On September 11, 2014, the study was reviewed and approved by the Independent Medical Ethics Committee of the ASL of Brindisi, the approval body responsible for health research conducted in the Province of Taranto (ID number: 142/14).

### 2.2. Study Design and Enrollment

Subject enrollment was conducted in primary schools. Within each of the three geographic zones delineated in the study, four public primary schools were selected, for a total of 12 schools (Table 1). These schools were chosen in order to accurately represent the environmental effects of industrial complex emissions within the Province of Taranto. To this effect, maps of average PM10 concentration at ground level in the Province of Taranto, created by the Puglia Regional Agency for Prevention and Protection of the Environment (ARPA Puglia) in 2010, informed school selection. After verifying children’s registration forms and eligibility for the study, participants were enrolled, and the cohort was balanced for zone, school, age, and gender. Data were anonymized by assigning each participant a numerical identification code.

### 2.3. Division of Residential Location into Residential Zones 

To categorize residential location, the Taranto municipality was divided geographically into three concentric residential zones (1, 2, 3) of increasing radial distance from the industrial complex. The industrial complex was defined as a polygonal region in which processes of steel preproduction (including iron ore extraction), production, and refinery were performed regularly (Figure 1). 

First, geolocalization of participants’ reported residence and school of attendance was performed using the World Geodetic System (WGS84). Thereafter, residential distance from the industrial complex was calculated for each participant as the straight-line distance between his/her home address and the closest point of the polygon denoting the perimeter of the complex. The perimeter, rather than center, of the industrial complex was used in distance measurement due to the presence of accumulated metal dusts at the industrial complex borders; wind forces have been known to resuspend these dusts and transport them airborne across the city. 

Subsequently, the distance from the industrial complex perimeter to the farthest participant residential location was measured and divided by three, to form three levels of geographical distance (zones). Residential distance from the industrial complex was categorized as follows: less than 5.000 km (Zone 1), between 5.000 km and 9.999 km (Zone 2), or between 10.000 km and 15.000 km (Zone 3).

### 2.4. Collection and Analysis of Fasting Blood Glucose Data 

Fasting blood glucose measurements were collected at participants’ primary schools by trained medical and nursing personnel of the Taranto ASL Department of Prevention. Venous whole blood was collected from participating children in the morning hours after fasting overnight, with at least one parent present. Blood specimens were drawn in heparinized tubes and promptly transported to the analytical laboratories of the Hospital of Taranto for immediate analysis. 

### 2.5. Collection of BMI Z-Score, SES, and Lifestyle Data

A written questionnaire adapted from the Elementary Home Observation for Measurement of the Environment (HOME) Inventory [17] was administered to the parent(s) of participants to obtain self-reported child data. For determination of BMI Z-score, these included participant birthdate, sex, height, and weight. Parental assessments of child height and weight have been described as reasonably accurate for this age group, particularly when classifying large cohorts into obese and non-obese categories [18,19].

Similarly, information on residential address, parental occupation, and parental education level was collected from parents via the Elementary HOME Inventory. Participant lifestyle data, including weekly hours of sports activity, weekly hours of outdoor play, and total weekly hours of physical activity, were also provided by parents in this manner. The questionnaire was completed by the parent(s) at the primary school of the child, during the evaluation of the child by study personnel.

### 2.6. Determination of BMI Z-Score

A clinical determination of underweight, healthy weight, overweight, or obese was made for each participant based on parent-reported data, using the World Health Organization (WHO) BMI-for-age reference charts for girls and boys. Based on these charts, participants with calculated BMI Z-scores below or equal to −2 standard deviations (SD) for their age and gender were classified as underweight. Participants with Z-scores from −2 SD up to and including +1 SD were classified as healthy weight. Z-scores from +1 SD up to and including +2 SD were considered overweight, and those greater than +2 SD were classified as obese [20,21]. 

To corroborate the accuracy of Z-score results, clinical weight classification via BMI was also conducted based on BMI percentile, using the Children’s BMI Tool for Schools of the United States Centers for Disease Control and Prevention (CDC; values not used for analysis). All correlations of BMI percentile and BMI-adjusted fasting blood glucose with other covariates remained at the same level of statistical significance using this approach. The distribution of participants into weight classifications was also comparable (6.5% underweight, 52.8% healthy weight, 20.1% overweight, 20.6% obese). 

### 2.7. Determination of SES

SES was classified for each participating child as low, medium, or high, according to the validated protocol described by Cesana et al [22]. The following characteristics of both parents were used as inputs to determine the SES classification of each child: age, years of formal education, occupation, and work-related stress perception. 

### 2.8. Statistical Analysis

The cohort was first described using standard statistics (mean and standard deviation for continuous variables; frequency, and percentage for discrete variables) to explore the characteristics of each residential zone and group. Generalized additive models were used to confirm a linear relationship between each independent variable—residential zone and SES—and each metabolic dependent variable of interest [23].

Linear modeling was then performed to explore these relationships quantitatively. Different residential proximities to the industrial complex were represented and compared via residential zone, the categorical variable described above (Zone 2 vs. Zone 1; Zone 3 vs. Zone 1). SES was expressed and compared in terms of the above-described categorized indices (High vs. Low; Medium vs. Low). We also performed a mediation analysis to test the role of BMI Z-score as a mediator in the association of SES and residential-to-industrial-complex distance with FBG. The 95% confidence intervals were calculated for each regression. Differences between groups were considered statistically significant at *p* < 0.05 for a two-tailed hypothesis test. All statistical analyses, tables, and graphs were performed and generated with R 3.5.2 [24].

## 3. Results

### 3.1. Sociodemographic and Physical Characteristics of the Cohort

The main sociodemographic and physical characteristics of participating children (*n* = 214) are summarized in Table 2. The three residential zones had similar numbers of participants (*n* = 62 for Zone 1; *n* = 77 for Zone 2; *n* = 75 for Zone 3). The cohorts were also well-balanced by sex (*n* = 114 or 53.3% female; *n* = 100 or 46.7% male). Zone 1 had more females than males, while Zones 2 and 3 had nearly equal numbers of males and females (62.9% female in Zone 1; 49.4% female in Zone 2; 49.3% female in Zone 3). The mean age of all study participants was 8.6 years, and the age distribution was quite consistent for all zones and sexes (8.9 ± 1.6 years for Zone 1; 8.2 ± 1.4 years for Zone 2; 8.6 ± 1.5 years for Zone 3; 8.5 ± 1.5 years for all females; 8.6 ± 1.5 years for all males). 

Across the three residential zones and both sexes, the distribution of height was quite uniform (133.6 ±11.5 cm for all participants). Weight was slightly higher in Zone 1 relative to Zones 2 and 3 (35.4 ± 10.9 kg for Zone 1; 31.5 ± 9.4 kg for Zone 2; 33.0 ± 10.0 kg for Zone 3), as well as in males relative to females (31.8 ± 9.0 kg for females; 34.7 ± 11.2 kg for males). 

In general, the BMI classification of participating children showed a disproportionately high prevalence of overweight and obesity (4.2% underweight, 50.9% healthy weight, 22.0% overweight, 22.9% obese). This trend towards overweight and obesity was especially strong among participants in Zone 1 (25.8% overweight, 27.4% obese) and among male participants (15% overweight, 36% obese), as over 50% of participants in these two groups were overweight or obese. Mean BMI Z-score for the cohort was 0.8 ± 1.5 (1.1 ± 1.5 for Zone 1; 0.62 ± 1.5 for Zone 2; 0.66 ± 1.6 for Zone 3; 0.56 ± 1.3 for females; 1.0 ± 1.7 for males).

Study participants tended towards low SES (42.5% low SES; 32.7% medium SES; 24.8% high SES). This tendency occurred because Zone 1 participants were disproportionately concentrated in the low SES index (66.1% low SES, 25.8% medium SES, 8.1% high SES). By contrast, Zone 2 and Zone 3 were relatively well-balanced by SES index (Zone 2: 33.8% low SES, 32.5% medium SES, 33.8% high SES; Zone 3: 32.0% low SES, 38.7% medium SES, 29.3% high SES). Further, males and females had SES distributions, with nearly equal portions of each sex in the low SES index, a slightly greater portion of females in the medium SES index, and a slightly greater portion of males in the high SES index (females: 42.1% low SES; 36.0% medium SES; 21.9% high SES; males: 43% low SES, 29% medium SES; 28% high SES). 

Fasting blood glucose values were within the age-appropriate range of normality for all subjects [25], with a cohort average of 85.5 ± 6.6 mg/dL, and are described in detail in Section 3.2.

### 3.2. FBG by Sociodemographic Factors

Table 3, Table 4 and Table 5 present regression coefficients and 95% confidence intervals for the association of sociodemographic factors with BMI Z-score and FBG (Table 3 describes all participants, Table 4 describes female participants only, and Table 5 describes male participants only). When analyzing FBG, the relationship between FBG and BMI Z-Score was taken into account. Overall, Zone 1 participants had significantly higher FBG compared to Zone 2 and Zone 3 participants (*p* < 0.05 vs. Zone 2; *p* < 0.001 vs. Zone 3; Table 3). When considering only female participants; Zone 1 continued to show significantly elevated FBG relative to Zone 3 (*p* < 0.001), with a higher magnitude of association (β_All participants_ = −4.6; β_Females_ = −6.3). The difference in FBG between Zones 1 and 2 dropped below the threshold of significance for female participants only, yet the magnitude of association increased relative to the full cohort (β_All participants_ = −2.5; β_Females_ = −3.0; Table 4). When only male participants were analyzed, none of the differences in FBG by residential zone remained statistically significant. For males only, FBG comparisons by residential zone showed lower magnitudes of association than in the full cohort (Table 5). Low SES participants did not show statistically significant differences in FBG relative to medium SES or high SES participants, neither for any particular sex nor for the cohort as a whole. 

BMI Z-score showed a positive association with FBG but was not statistically significant for any sex or for the cohort as a whole. The same analysis was performed using categorical BMI, and results were very similar to those previously described. Examining FBG results by clinical BMI subgroup (with all participants included), we found an almost significant difference in FBG between healthy weight subjects and obese subjects, with the former category showing a lower average FBG concentration (β = −2.0; CI = −4.2, 0.3). 

Additionally, in our analyses, BMI Z-score was not a mediator of the relationships between FBG and either residential zone or SES (results not shown). When FBG was not corrected for BMI Z-score, similarly significant associations at the same *p*-value thresholds were found between these residential zones and SES indices.

### 3.3. BMI Z-Scores and Clinical Subgroups by Sociodemographic Factors

Results on the impact of sociodemographic factors on BMI Z-score are shown in Table 3, Table 4 and Table 5. No statistically significant relationship was found between residential zone and BMI Z-score for any sex or for the cohort as a whole. BMI Z-score was also not significantly different across SES categories for males, females, or the entire cohort. 

Across the cohort, participants whose mothers had completed high school education (13 years of schooling) had significantly lower BMI Z-scores than those whose mothers were formally educated up to middle or primary school (5–8 years) (*p* < 0.05; Table 3). The same was true for participants whose mothers had a university-level education or higher (16+ years in total) with respect to those whose mothers completed up to middle or primary school (*p* < 0.05; Table 3). Likewise, female children whose mothers completed high school had significantly lower BMI Z-scores than female children whose mothers completed only 5–8 years of formal education (p < 0.05; Table 4). However, the relationship between BMI Z-score and maternal education level was no longer significant when comparing female children whose mothers had university-level education or higher with female children whose mothers had only middle or primary school education (Table 4). For male children, no comparison by maternal education level showed a significant difference in BMI Z-score (Table 5). 

Furthermore, no significant association with BMI Z-score or FBG was found for weekly hours of sports activity, weekly hours of outdoor play, or total weekly hours of physical activity (*p* > 0.05 for all comparisons). Similarly, no significant difference between residential zones was found for any of the lifestyle variables (*p* > 0.05 for all comparisons). Low SES participants had significantly lower reported weekly hours of outdoor play than high SES participants (*p* < 0.05), although all other comparisons of lifestyle outcomes by SES were statistically nonsignificant.

## 4. Discussion

This study describes the BMI values and FBG levels of a cohort of 214 Taranto schoolchildren aged 6–11 years, in relation to residential distance from a nearby industrial complex and SES. Our results showed that residential proximity to the complex was associated with high BMI-adjusted FBG. This effect of residential location on BMI-adjusted FBG was stronger in female children than in male children. Residential proximity to the industrial complex was not associated with BMI Z-score, and SES was not associated with BMI Z-score or FBG. Additionally, higher maternal education level was associated with lower BMI Z-score. Both BMI Z-score and the resultant clinical subgroups (underweight, healthy weight, overweight, obese) showed positive but nonsignificant associations to FBG.

### 4.1. Fasting Blood Glucose Concentration by Sociodemographic Factors

Despite no significant difference in BMI Z-score makeup across SES and residential groups, FBG was higher in children who lived closer to the industrial complex. At the same time, SES and BMI Z-score were not associated with FBG. Taken together, these results show that residential location with respect to the industrial complex influenced children’s FBG levels; they suggest that the presence of the industrial complex plays a role in the corporeally elevated fasting blood glucose of children in the surrounding area. While consistent with preliminary literature on the child health outcomes and hyperglycemia in the setting of industrial metal exposure [26,27,28,29], this association has not been described previously. More thorough investigation of the relationship between residential proximity to industrial activity and increase in pediatric FBG is highly merited. 

As residential proximity to the industrial complex increased, female children experienced substantially more significant increases in FBG than male children. This male–female disparity was strongly apparent in the Zone 1 vs. Zone 3 comparison. The Zone 1 vs. Zone 2 comparison also suggested that females may be more vulnerable to increased FBG based on closer residential location to the industrial complex, yet the reduced sample size of the single-sex analyses limited the detection of statistically significant differences between boys and girls at this level. These findings concur with prior literature that describes female children as potentially more vulnerable than male children to the health consequences of environmental exposure [30,31]. For instance, a study of 6730 Chinese children aged 3–7 years found that higher concentrations of particulate matter, sulfur dioxide, and nitrogen dioxides in ambient air were significantly associated with increased respiratory symptoms in girls, but not in boys [31]. However, since there is no clear consensus about the effects of sex on pollution sensitivity, nor about the reasons for which pollution sensitivity may differ between boys and girls, the role of sex in the relationship between FBG and residential proximity to the industrial complex should be studied further. In addition, further analysis is required to understand the mechanisms of this effect. For instance, FBG levels could be mediated in several ways by environmental exposure to obesogenic and metabolically disruptive steel production byproducts. Potential pathways for this mechanism include ambient air or water contamination, a greater frequency of contact with persons directly exposed to toxins from the industrial complex, or behavior modification due to negative perceptions of the industrial complex. Considering the lack of a statistically significant BMI Z-score gradient with respect to residential proximity to the industrial complex, the latter possibility seems less probable. 

Moreover, these data suggest a geographic threshold for the negative effects of residing near an industrial complex. While Zone 1 had significantly worse outcomes than Zones 2 and 3 with regard to FBG levels, differences between Zones 2 and 3 were not statistically significant. This finding indicates that a categorical interpretation of distance may be effective for identifying groups most impacted by industrial activity. 

The positive association of BMI Z-score with FBG and the nearly significant difference in FBG between healthy weight and obese participants may be relevant to preventative health measures. A 2017 study of obese, nondiabetic children aged 8–15 years in the United States found that proximity to air with nitrogen dioxide and particulate matter contaminants decreased insulin sensitivity and increased Type II diabetes risk at early adulthood [32]. The interaction between BMI status, FBG, and environmental exposure suggested by our study and others should be further explored, in order to inform clinical and public health management of metabolic disease risks in childhood and early adulthood.

### 4.2. BMI Z-Scores and Clinical Subgroups by Sociodemographic Factors

As a whole, the BMI Z-score makeup of participating children was congruent with that described by prior studies of age-matched Italian cohorts; 44.9% of children were overweight or obese, and higher obesity and overweight/obesity rates were found in male children than in female children [4,5,6,7,8,9,10]. In our study, overweight or obese status were not linked to any particular socioeconomic or residential condition in a statistically significant way. This contrasts with prior literature, in which the presence of overweight/obesity in children was linked to economic disadvantage [33,34,35] and poor neighborhood conditions [36,37]. However, the comparatively smaller size of the Taranto cohort may have limited the statistical power of our study to capture SES influences.

The association of higher maternal education level with lower pediatric BMI Z-score, and the lack thereof with BMI-adjusted FBG, agrees with previous literature [5,8]. In adults, formal education level has been shown to affect health literacy and, in turn, health outcomes [38]. Given the central role of the maternal figure in controlling child diet in Southern Italian society, the influence of maternal education level on child BMI was considered worthy of investigation. Notably, maternal education was lowest in Zone 1, the area closest to the plant and with the lowest overall SES. This difference in maternal education by zone would likely have been more apparent with a larger sample size. 

Similarly, the weakening of the association between maternal educational level and BMI Z-score when the cohort was disaggregated by sex most likely results from the reduction in sample size (*n* = 214 total participants; *n* = 114 females, *n* = 100 males). In 2019, a population-based study of 45,000 Italian adolescents recorded significantly lower overweight/obesity rates for both males and females as maternal education increased [39]. In our study, however, the stronger relationship between maternal education and BMI Z-score for female children relative to male children is nonetheless noteworthy. This observation merits deeper investigation, particularly in light of the more intense association of residential location to BMI-adjusted FBG for girls. 

Additionally, the association between maternal education level and child BMI Z-score supports the notion that obesity is mediated by the interaction of multiple different environmental conditions, rather than a single predominant factor. A genetic modeling study of 925 pediatric twin pairs found that children living in obesogenic homes (defined by quantitative measurement of the food, physical activity, and media influences present in the household) displayed greater BMI heritability than those in non-obesogenic homes [40]. Such literature suggests that the home environment in early life may play an important role in modifying genetic propensities for obesity.

In this regard, the lack of association between BMI Z-score and weekly hours of sports activity, weekly hours of outdoor play, or total weekly hours of physical activity in this study is notable. This result suggests that, while an important component of obesity prevention, physical activity alone may be insufficient to alter obesity prevalence on a population level. This insight concords with past literature and is an important consideration for public health policies and programs addressing obesity [41,42].

Finally, the SES index used in our analyses was derived from the parental occupation and education level. As a result, it may have been differently informative about parental awareness of, and ability to afford, healthy lifestyle options when compared to the SES classifications used by other studies [5,33,34,35]. For instance, the idea that parental occupation may not be related to childhood overweight and/or obesity is corroborated by prior work [5]. Finally, it is possible that obesogenic factors originating from the industrial complex are ubiquitous in the local environment and affect the entire cohort, altering expected patterns in overweight/obesity [5,36,37].

### 4.3. Implications

This study found that closer residential location to a nearby industrial complex on the order of 5.000 km was significantly associated with worsened metabolic outcomes. This result is consistent with past studies conducted by our research group in Brescia Province (Italy), in which greater outdoor dust Manganese concentrations were observed within 2.000 km of ferromanganese alloy plants [43]. SES (low, medium, or high) was not significantly associated with any studied change in metabolic variables. Therefore, these findings support the notion that residential area may be a more effective starting point to reach populations experiencing the negative health effects of industrial exposure [44,45]. Furthermore, the FBG consequences of residing near the industrial complex were more pronounced in female children than in male children. This indicates that young girls may be more susceptible to the health effects of industrial exposure, and that their potentially elevated health risks from exposure merit special attention.

### 4.4. Strengths, Limitations, and Future Directions

This study describes the interrelationships between two important environmental factors (SES and residential location) and the metabolic health outcomes of a unique, sizable, and robust pediatric cohort. While previous publications have focused on urban children aged 6–11 years or populations residing near industrial sites, few studies have explored both conditions simultaneously. The quantity and diversity of available data allowed us to study the metabolic results of BMI and FBG together. For this reason, a fuller understanding of how SES and residential proximity to the industrial complex influence metabolic health could be obtained. This cohort was also balanced across the variables of age, sex, SES, and residential proximity to the industrial complex, offering a useful system to study these multiple relationships.

However, this study was not without limitations. BMI inputs of height and weight were based on parental perceptions and not measured according to a standardized protocol, allowing for subjective bias that is difficult to quantify. Furthermore, the sample size of this cohort did not provide enough statistical power to analyze severely obese participants in a separate BMI category. Severe obesity is defined as a BMI percentile value over 1.2 times that of the 95th percentile for the child’s age and gender according to the WHO reference curve and predicts distinctly elevated cardio-metabolic risk [46].

Another important consideration regards the approach by which height, weight, age, and sex data are converted into pediatric BMI measurements and cutoffs for underweight, healthy weight, overweight, and obese. The WHO standards used in this paper are considered most appropriate for measuring obesity and overweight at the population level, given their higher sensitivity relative to other systems. Additionally, the use of Z-scores was determined to be more statistically meaningful for our analytical purposes than the percentile values generated by the CDC tool [20,21,46]. At the same time, Italian national standards for pediatric BMI measurement and classification have also been developed previously. While more specific than the WHO standards used herein for identifying children at high cardiovascular risk, these standards are considered less useful for cohort studies due to their lower sensitivity and tendency to underestimate rates of overweight and obesity in larger samples [47]. The analysis and interpretation of pediatric BMI values in this study is advantageous for the sensitivity and broad applicability of the study, although other methods to ensure compatibility with national and regional tendencies in body composition should also be considered.

Participants’ residential distance from the plant was analyzed as a categorical variable in order to represent the neighborhood structure of the Taranto municipality. In this way, the study design attempted to maintain consistency with health policies and interventions, which are most often organized on a discrete zone-by-zone or neighborhood-by-neighborhood basis. At the same time, two potential limitations of this approach exist. 

First, the similarity and contrast of environmental exposure profiles within and across the 5.0 km zones, respectively, have not been experimentally verified. Second, therefore, multiple equally reasonable zone delineations might have emerged when clustering participant residences by distance from the industrial complex. However, when the same correlations were run using distance as a continuous variable (measured in kilometers from the perimeter of the industrial complex), nearly identical correlations were obtained. This suggests that categorical measures of distance employed herein are an appropriate unit of analysis in the context of this study.

These results would also be stronger if combined with quantitative measures of industrial complex-associated pollution in the studied residential areas and/or with biomarkers of exposure. Therefore, future studies from our group will aim to measure industrial particles and byproducts in ambient air, soil, and water of the three residential zones, as well as participants’ physiological uptake of these substances [48]. By specifying the relationship between residential location and environmental/physiological exposure to industrial activities, these data will better contextualize the results described herein. Additionally, expanding this cohort to include a control population far removed from the industrial complex and accounting for variation in participants’ dietary and lifestyle behaviors will permit a more potent interpretation of these results [49].

More broadly, sociodemographic and biological factors beyond the scope of this study such as home environment and diet may play relevant obesogenic roles in interaction with SES and residential proximity to the industrial complex [50]. Other biological variables should also be incorporated into future studies. For instance, blood creatinine levels were collected for all participants in this cohort. These data were not analyzed in relation herein, because biological and environmental mediators of blood creatinine levels are vast. However, these data merit further investigation. In fact, prior studies have demonstrated a link between obesity and elevated serum creatinine levels in children above 10 years of age, with implications for renal function [51]. In a prior cohort study of 396 Italian children aged 5–11 years, urine creatinine levels were directly correlated with measurements of urban benzene pollution surrounding the home environment [52]. Therefore, more detailed and multifactorial analyses are required to elucidate the relationships of residential location and SES with these variables.

## 5. Conclusions

Our study found that greater residential proximity to an industrial complex was associated with significant increases in FBG concentration among a cohort of primary school children in Taranto, Italy. BMI Z-scores were not significantly different by residential zone, and neither BMI Z-score nor FBG was significantly associated with SES. Moreover, the relationship between decreased residential distance from the industrial complex and increased FBG was particularly strong among female children. Taken together, these results provide additional evidence that living close to industrial activity can negatively affect the metabolic health of children aged 6–11 years. These data also support residential location as a fundamental consideration for reaching the pediatric populations most acutely impacted by industrial exposure. They corroborate the need for customized interventions that address potential differences in the health needs and risk profiles of boys and girls. As a result, this study provides useful evidence for the design of public health policies and interventions for children living in the setting of industrial activity.

## Figures and Tables

**Figure 1 ijerph-16-02036-f001:**
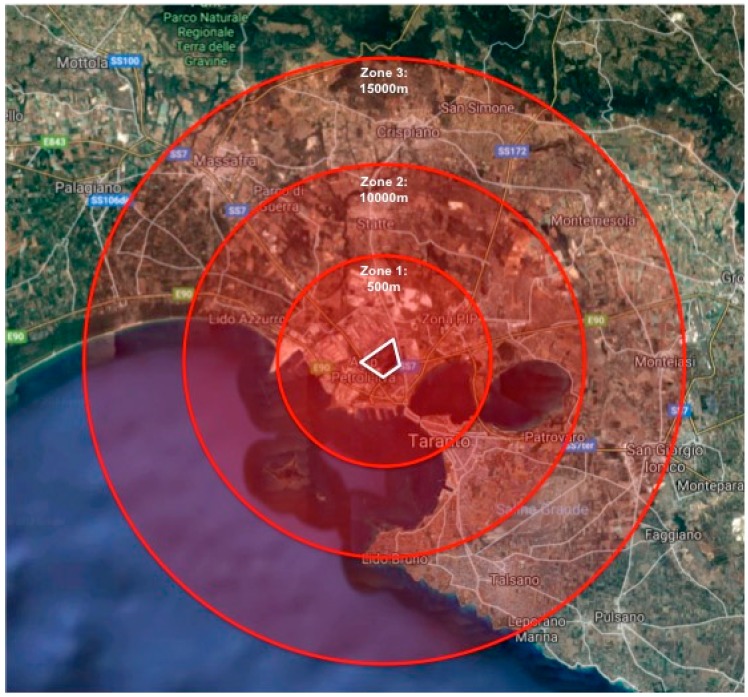
Geographic description of the residential zones (circles) used to define participant proximity to the industrial complex (polygon).

**Table 1 ijerph-16-02036-t001:** Description of primary school enrollment sites by residential zone, distance from industrial complex, and neighborhood.

Residential Zone	Radial Distance from Industrial Complex (km)	Neighborhoods	Primary Schools
1	0.000–4.999	Tamburi, Città Vecchia, Borgo	Vico, Deledda, Giusti, XXV Luglio
2	5.000–9.999	Italia Montegranaro, Tre Carrare Battisti, Solito Corvisea, Salinella, Paolo VI, Comune di Statte	Alfieri, Viola, Pertini, Giovanni XXIII
3	10.000–15.000	Talsano-San Vito-Lama	Frascolla, Salvemini, Sciascia, De Amicis

**Table 2 ijerph-16-02036-t002:** Sociodemographic and physical characteristics of the study cohort (mean (± SD) or *n* (%)) in totality, by residential zone, and by sex (BMI = Body mass index; SES = socio-economic status; FBG = fasting blood glucose).

Measured Characteristic	Total (*n* = 214)	Zone 1 (*n* = 62)	Zone 2 (*n* = 77)	Zone 3 (*n* = 75)	Females (*n* = 114)	Males (*n* = 100)
**Sex (F)**	114 (53.3%)	39 (62.9%)	38 (49.4%)	37 (49.3%)		
**Age (years)**	8.6 (±1.5)	8.9 (±1.6)	8.2 (±1.4)	8.6 (±1.5)	8.5 (±1.5)	8.6 (±1.5)
**Weight (kg)**	33.1 (±10.1)	35.4 (±10.9)	31.5 (±9.4)	33.0 (±10.0)	31.8 (±9.0)	34.7 (±11.2)
**Height (cm)**	133.6 (±11.5)	134.5 (±13.5)	132.3 (±10.6)	134.0 (±10.8)	132.9 (±11.2)	134.3 (±11.9)
**BMI Z-Score**	0.8 (±1.5)	1.1 (±1.5)	0.62 (±1.5)	0.66 (±1.6)	0.56 (±1.3)	1.0 (±1.7)
Underweight	9 (4.2%)	2 (3.2%)	5 (6.5%)	2 (2.7%)	5 (4.4%)	4 (4%)
Healthy Weight	109 (50.9%)	27 (43.6%)	39 (50.6%)	43 (57.3%)	64 (56.1%)	45 (45%)
Overweight	47 (22.0%)	16 (25.8%)	20 (26.0%)	11 (14.7%)	32 (28.1%)	15 (15%)
Obese	49 (22.9%)	17 (27.4%)	13 (16.9%)	19 (25.3%)	13 (11.4%)	36 (36%)
**SES**						
Low	91 (42.5%)	41 (66.1%)	26 (33.8%)	24 (32.0%)	48 (42.1%)	43 (43%)
Medium	70 (32.7%)	16 (25.8%)	25 (32.5%)	29 (38.7%)	41 (36.0%)	29 (29%)
High	53 (24.8%)	5 (8.1%)	26 (33.8%)	22 (29.3%)	25 (21.9%)	28 (28%)
**FBG (mg/dL)** ^1^	85.5 (±6.6)	87.8 (±6.3)	85.5 (±7.1)	83.5 (±5.6)	85.4 (±6.8)	85.6 (±6.3)

^1^ For FBG data, *n* = 212 for cohort (Zone 1: *n* = 62; Zone 2: *n* = 76; Zone 3: *n* = 74; females: *n* = 113; males: *n* = 99).

**Table 3 ijerph-16-02036-t003:** All participants (*n* = 214): Linear regression coefficients and 95% confidence intervals for associations of (i) residential zone; (ii) socioeconomic status (SES) index; and (iii) maternal education level to body mass index (BMI) Z-score and BMI-adjusted fasting blood glucose (FBG) (mg/dL).

Sociodemographic Variable	BMI Z-Score	FBG Adjusted by BMI Z-Score
**BMI Z-Score**		0.337
		(−0.254, 0.928)
**Zone 2 vs. Zone 1**	−0.223	−2.518 *
	(−0.755, 0.309)	(−4.813, −0.224)
**Zone 3 vs. Zone 1**	−0.052	−4.638 ***
	(−0.606, 0.501)	(−7.020, −2.255)
**SES: Medium vs. Low**	−0.23	1.453
	(−0.729, 0.269)	(−0.699, 3.606)
**SES: High vs. Low**	0.199	1.71
	(−0.699, 1.096)	(−2.157, 5.576)
**Maternal Education: 13 years vs. 5–8 years**	−0.539 * (−1.056, 0.023)	−0.203 (−2.450, 2.044)
**Maternal Education: 16+ years vs. 5–8 years**	−1.031 * (−2.014, −0.048)	−0.761 (−5.037, 3.514)
**Constant**	1.347 ***	87.036 ***
	(0.934, 1.760)	(85.089, 88.983)

Legend: * *p* < 0.05; *** *p* < 0.001.

**Table 4 ijerph-16-02036-t004:** Female participants (*n* = 114): Linear regression coefficients and 95% confidence intervals for associations of (i) residential zone; (ii) socio-economic status (SES) index; and (iii) maternal education level to body mass index (BMI) Z-score and BMI-adjusted fasting blood glucose (FBG).

Sociodemographic Variable	BMI Z-Score	FBG Adjusted by BMI Z-Score
**BMI Z-Score**		0.537
		(−0.412, 1.485)
**Zone 2 vs. Zone 1**	−0.011	−3.033
	(−0.629, 0.608)	(−6.080, 0.015)
**Zone 3 vs. Zone 1**	0.015	−6.312 ***
	(−0.676, 0.706)	(−9.717, −2.907)
**SES: Medium vs. Low**	−0.209	0.659
	(−0.825, 0.408)	(−2.383, 3.702)
**SES: High vs. Low**	−0.002	3.623
	(−1.406, 1.402)	(−3.291, 10.536)
**Maternal Education: 13 years vs. 5–8 years**	−0.664 * (−1.279, −0.048)	1.037 (−2.059, 4.132)
**Maternal Education: 16+ years vs. 5–8 years**	−0.537 (−2.015, 0.942)	−2.899 (−10.199, 4.401)
**Constant**	0.998 ***	87.319 ***
	(0.527, 1.469)	(84.813, 89.825)

Legend: * *p* < 0.05; *** *p* < 0.001.

**Table 5 ijerph-16-02036-t005:** Male participants (*n* = 100): Linear regression coefficients and 95% confidence intervals for associations of (i) residential zone; (ii) socio-economic status (SES) index; and (iii) maternal education level to body mass index (BMI) Z-score and BMI-adjusted fasting blood glucose (FBG).

Sociodemographic Variable	BMI Z-Score	FBG Adjusted by BMI Z-Score
**BMI Z-Score**		0.265
		(−0.536, 1.066)
**Zone 2 vs. Zone 1**	−0.664	−1.213
	(−1.617, 0.289)	(-4.918, 2.491)
**Zone 3 vs. Zone 1**	−0.458	−1.659
	(−1.414, 0.498)	(−5.354, 2.036)
**SES: Medium vs. Low**	−0.104	2.994
	(−0.945, 0.737)	(−0.241, 6.229)
**SES: High vs. Low**	0.196	1.303
	(−1.090, 1.483)	(−3.649, 6.255)
**Maternal Education: 13 years vs. 5–8 years**	−0.462 (−1.344, 0.419)	−1.959 (−5.369, 1.452)
**Maternal Education: 16+ years vs. 5–8 years**	−1.311 (−2.788, 0.166)	0.196 (−5.582, 5.974)
**Constant**	1.923 ***	85.993 ***
	(1.173, 2.674)	(82.721, 89.266)

Legend: *** *p* < 0.001.

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
