# Peer review of "Metabolic Outcomes in Southern Italian Preadolescents Residing Near an Industrial Complex: The Role of Residential Location and Socioeconomic Status"

_ijerph, 2019, doi:10.3390/ijerph16112036_

Round 1

Reviewer 1 Report

1.   The standard of English needs to be improved in this manuscript.

2.   There are old references that can be replaced with more recent ones.

3. Line 33: Do you mean fasting blood glucose? Please clarify throughout.

4.   Line 47-48: Some keywords are inappropriate- please revise. Suggest delete metabolism, environmental health and population; and include either obesity or BMI.

5.   In introduction, the significance and implications of the study are not sufficiently addressed. Many sentences are not clear and concise (e.g. Line 55-56).  

6. Line 115-118: This paragraph is almost unclear and should be enlarged. Please clarify why public schools were only selected? What is meant by screening questionnaires?

7.   Line 142-143: The classification is not clear here.

8. Line 152: This section is not detailed enough. How many written questionnaires? How reliable are questionnaires in this study?

9. Line 178-180: Please clarify how the data were analyzed using Generalized additive models (GAMs)?

Author Response

Dear Reviewer,

    Thank you for your generous, critical feedback on our manuscript. The authors very much appreciate the provided suggestions, which we found both insightful and extremely helpful, as well as the opportunity to revise and improve our article.

    Our study team has thoroughly reviewed the comments and updated the manuscript to reflect all suggested changes. In the attached letter, we have included the comments and responded to each of them in a point-by-point manner (in red text), explaining how we have resolved each concern in the manuscript. In the revised manuscript submission, all changes made have been marked via the Track Changes function.

    We are hopeful that the revised manuscript is now suitable for publication in the International Journal of Environmental Research and Public Health, and we look forward to hearing from the editorial board.

Sincerely and on behalf of all authors,

Esha Bansal

MD/MPH Candidate

Icahn School of Medicine at Mount Sinai

1 Gustave L. Levy Place

New York, NY 10029 (USA)

321-537-0601

esha.bansal@icahn.mssm.edu

Reviewer 2 Report

This is a very interesting study, and in my opinion it should be published. before publication however, few modifications of the manuscript are necessary.

Please give more information regarding recruitment of the participants. I found no statement regarding ethic approval.

 I would recommend to analyze male and female children separately.

Please present socidemographic and physical characteristics for each sex separately in table 1. Maybe girls and boys differ in their sensitivity to industrial pollution

Author Response

Dear Reviewer,

    Thank you for your kind words and valuable, critical feedback on our manuscript. The authors very much appreciate the provided suggestions, which we found both insightful and extremely helpful, as well as the opportunity to revise and improve our article.

    Our study team has thoroughly reviewed the comments and updated the manuscript to reflect all suggested changes. In the attached letter, we have included the comments and responded to each of them in a point-by-point manner (in red text), explaining how we have resolved each concern in the manuscript. In the revised manuscript submission, all changes made have been marked via the Track Changes function.

    We are hopeful that the revised manuscript is now suitable for publication in the International Journal of Environmental Research and Public Health, and we look forward to hearing from the editorial board.

Sincerely and on behalf of all authors,

Esha Bansal

MD/MPH Candidate

Icahn School of Medicine at Mount Sinai

1 Gustave L. Levy Place

New York, NY 10029 (USA)

321-537-0601

esha.bansal@icahn.mssm.edu

Round 2

Reviewer 1 Report

No Further comments.

Reviewer 2 Report

In my opinion the manusript is now ready for publication